# A Study on the Role of Pre-Cleaning and a New Method to Strengthen Gate Oxide Quality

**DOI:** 10.3390/nano12091563

**Published:** 2022-05-05

**Authors:** In-Kyum Lee, Byoung-Deog Choi

**Affiliations:** 1Department of Semiconductor and Display Engineering, Sungkyunkwan University, 2066 Seobu-ro, Jangan-gu, Suwon 16419, Korea; brienlee@g.skku.edu; 2Department of Electrical and Computer Engineering, Sungkyunkwan University, 2066 Seobu-ro, Jangan-gu, Suwon 16419, Korea

**Keywords:** IPA, O_3_ deionized water, gate oxide, breakdown

## Abstract

Isopropyl alcohol (IPA) has been conventionally used for pre-cleaning processes. As the device size decreased, the gate oxide layer became thinner. As a result, the quality of the gate oxide was degraded by a pre-cleaning process, and oxide reliabilities and product yield were affected. In this study, we investigate whether the carbon generated on the silicon interface after the IPA drying process might have induced gate oxide breakdown. Time-dependent dielectric breakdown (TDDB) failure increased in frequency since carbon contaminations were increased in the oxide according to the amount of IPA. Organic contaminations resulted in a lower energy level, and electron tunneling occurred through the gate oxide. When an external electric field was applied, organic materials in the gate oxide layer were aligned, and a percolation path formed to cause breakdown. Finally, we suggest a new cleaning method using carbon-free O_3_ deionized (DI) water as a dry-cleaning method to improve oxide dielectric breakdown. An O_3_ DI dry cleaning process could reduce carbon particles in the oxide layer and decrease gate oxide failure by 7%.

## 1. Introduction

The gate oxide is one of the most important features of MOSFET semiconductors. For instance, Figure 1a shows how the demand for DRAM products has changed over the years; PC DRAM shipments have been decreasing. However, server DRAM shipments have shown consistent growth over the past years, which reached 33 percent of worldwide DRAM shipments in 2020, up from 11 percent in 2005 [1]. For the long-term maintenance of data centers, it is necessary to manufacture a chip with high endurance of more than ten years that is different from existing products. Moreover, as development progresses, the driving voltage must be lowered to reduce thermal and power consumption according to the increased density and frequency. Figure 1b shows the decrease in driving voltage according to the development of DRAM, for which the Joint Electron Device Engineering Council (JEDEC) published a corresponding standard [2]. The operating voltage was lowered to reduce power consumption while the performance was improved. As a result, the gate oxide layer was thinner. Sub-10 nm Complementary Metal-Oxide Semiconductor (CMOS) technology requires the use of ultra-thin gate oxides to compensate for some shortcomings, such as the short-channel effect, and to ensure device drive function and correct operation [3]. However, several disadvantages have been observed. One of the biggest concerns is the rapid increase in the continuous tunneling current across the gate oxide under circuit operation [4]. Figure 1c indicates the mechanism of increasing the electric field between the gate and silicon, which means that the probability of gate oxide breakdown is higher because as the oxide thickness becomes thinner, the electric field increases. As the electric field increases, the breakdown phenomenon appears more easily.

The dielectric breakdown is a general phenomenon in insulators [5] and is related to nanoscale defects, such as organic materials, particle, metallic impurities or precipitates, which jeopardize adequate semiconductor lifetime. Dielectric breakdown can be induced in a mass production manufacturing technology systems. Figure 2a shows many oxide charge types. There are four different charges within an oxide and at the Si interface. Within the oxide, there are mobile ionic charges, which have sufficient mobility to drift in the oxide under an applied electric field and trapped charges, which are high-energy electrons or holes injected into the oxide. At the interface, there are fixed and trapped charges such as dangling bonds. Figure 2b describes charges before the external voltage is applied, wherein these charges move around randomly. However, these charges make a conducting path, which is also a percolation path, and which finally results in insulator breakdown, as shown in Figure 2c. This event occurs due to the buildup in bulk of charges or defects produced by stress at the gate voltage.

This phenomenon is widely studied due to oxide breakdown from the accumulation of point-by-point defects, and is facilitated by the movement of carriers through the dielectric. Consequently, defects are accumulated in the insulator or interface between Si and SiO_2_ [6]. The effect is caused by the formation of a percolation path between the gate and silicon because of sub-1 nm defects [7]. Because of this conducting path, the drain current is no longer controlled by the gate voltage control; as a result, CMOS lose its characteristics. Specifically, the gate oxide breakdown phenomenon cannot be detected immediately during the manufacturing process, so it takes a long time to improve the oxide quality, which can cause huge cost loss. In this paper, we study the influence of carbon among the mechanisms by which defects caused by various factors affect the deterioration of the gate insulator. During the manufacturing process, it is important to reduce carbon adsorption on the Si substrate before the gate oxide is formed. If not, device reliability, as well as system performance, can be seriously affected [8]. Therefore, the pre-cleaning process, which can directly affect the surface condition of the Si/oxide interface and the removal of pollutants, has close correlation with oxide reliability. Wet chemical cleaning processes are essential to remove contaminants, and due to the rapid decrease in pattern size, are becoming more important than ever. However, it is well known that when a silicon surface, which is hydrophobic, is dried in a wet cleaning process, watermarks are easily created. Due to this problem, IPA dry technology has been utilized to prevent watermark formation [9]. Furthermore, by using the unique hydrophilicity and high permeability characteristics of IPA, the final rinse-completed wafer is put into the IPA vapor to remove particles remaining on the wafer completely, and at the same time, utilize the high volatility characteristic of IPA for complete drying [10]. However, in a recent study, organic components remaining on the Si surface both interfere with the formation of a uniform structure inside the gate oxide, and form a direct conduction path due to high electrical conductivity. So, the breakdown of the oxide film is a major cause of quality constraints [11,12]. The purpose of this study is to present a new solution to improve gate oxide quality by identifying the change in carbon content due to IPA cleaning, and determining the relationship to gate oxide reliability.

## 2. Experiments and Methods

To verify the effect of the IPA experiments that were conducted in a mass-production sub-20 nm DRAM device. Figure 3 shows the experimental process used to verify the influence of pre-cleaning on the gate oxide. First, a 60 Å-thick oxide layer was deposited on the silicon substrate using atomic layer deposition (ALD) (Figure 3a). Thereafter, the initial oxide thickness of 60 Å was removed by submerging in 1:30 HF: DI water for 40 s (Figure 3b). The initial ALD deposited SiO_2_ had been used as a sacrificial layer to protect the Si from various processes such as implant and etch. However, this film was compromised by a lot of manufacturing processes such as implants, photos and etch. So, it must be removed, and then the new oxide must be regrown to play a crucial role as an insulator. Moreover, the native oxide was removed by applying diluted HF (DHF) 1:200 HF: DI water for 85 s and then rinsing with deionized water (DIW) (Figure 3c). Experiment 1 was fabricated with different IPA concentrations of (4.1, 5.4, and 10.8) ppb for Exp-A, B, and C, respectively. The wafer drying process was carried out for 30 s using N_2_ gas after the cleaning process was accomplished. The gate oxide was grown to 60 Å in a furnace to form an insulator (Figure 3d). The oxide thickness was measured by randomly selecting samples from among the produced wafers. The thickness was measured by repeating 13 different positions on the wafer for each sample using thickness measuring equipment. Experiment 2 was undertaken with the fabricated dry method, replacing IPA with O_3_ DI to reduce organic particles caused by the IPA.

## 3. Results & Discussion

Figure 4a shows the carbon-to-carbon bonding spectrum analyzed by X-ray photoelectron spectroscopy (XPS, Thermo Fisher Scientific, ESCALAB 250, Seoul, Korea). It seems that as the IPA concentration increased, the number of carbon particles remaining at the silicon interface was increased. It is expected that the remaining carbon particles at the interface acted as a conducting trap in the subsequent oxide growing process. Figure 4b shows Weibull distribution curves for the breakdown voltage. The experimental values collected for the breakdown voltage were subjected to Weibull statistical analysis. The following formula is the Weibull cumulative distribution function [13]:(1)$$F\left(V\right)=1-\exp\left[-{\left(\frac{V-\gamma }{\alpha }\right)}^{\beta }\right]$$
where, *β* is the shape parameter, *α* is the scale parameter, and *γ* is the location parameter. The Weibull fit only considers *β* and *α*, and *γ* is zero to consider the breakdown voltage characteristic. Table 1 shows the Weibull scale and shape parameters.

To verify the long-term reliability, we conducted a time-dependent dielectric breakdown (TDDB) test. In the state where the substrate and source are set to ground, we applied −0.5 V to the drain and 5 V to the gate as step pulses at three seconds intervals to confirm the oxide degradation under repetitive voltage stress higher than the operating voltage. After that, the current was measured and the number of devices exceeding 500 µA of IPP current was counted. Samples with oxide breakdown tended to go out of the normal IPP current range from (300 to 480) µA. Moreover, the current flowed over 500 µA, so the limit was selected. We repeated the test using more than 10,000 samples for each of the experiment. Figure 5 describes the IPA concentration (line) and TDDB failure rate (bars). As the concentration of the IPA increased, the TDDB failure rate also increased. It was found that the amount of IPA drying after a wet cleaning process affected the quality of the gate oxide.

As the hydrogen fluoride (HF) removes the initial oxide, fluorine which is chemically stable, combines with Si to form the f-termination state shown in Figure 6a. Subsequently, the reaction occurs once more with the HF molecules by which polarization is generated because of the difference in electronegativity between Si and F, which were 1.9 and 3.8, respectively, as shown in Figure 6b,c. As a consequence, the Si surface changes from F-termination to H-termination, while the fluorine which become silicon tetra-fluoride (SiF) is removed from the Si interface, as shown in Figure 6d. Some of the h-termination-formed on the Si surface became OH terminated when it was exposed to moisture during the process, as shown in Figure 6e.

A higher IPA concentration increases the opportunity for hydrogen to bond with IPA and generates the OH-terminated silicon surface, as shown in Figure 7a. As SiO_2_ is regrown on the wafer, carbons composed of IPA try to bond with the Si-O compound, as shown in Figure 7b. Lastly, Figure 7c shows that it would result in a lower-energy band level. Therefore, when an external electric field is applied, electrons in the Si area easily move to the gate by direct tunneling. Moreover, these electrons have sufficient energy to generate an electron-hole pair (EHP). Some holes move to Si through SiO_2_, and create another EHP, while the others are trapped in the SiO_2_, which may increase the leakage of current caused by gate-oxide breakdown. When the moisture increases according to the humidity of each facility during IPA drying, the residual properties on the Si surface of the IPA molecules were raised [14]. This phenomenon is due to water molecules in the atmosphere forming a strong hydrogen bond between the IPA and Si surfaces, which have a small polarity, and act as a linker to connect the particles. Therefore, to prevent gate-oxide breakdowns, both the amount of IPA should be controlled, and the moisture inside the process chamber should be removed.

Previously, as a method to reduce carbon particle contamination by the IPA drying process, we conducted another experiment in which ozone-deionized water (O_3_ DI) was utilized. Exp-D for the IPA drying process and Exp-E for the new O_3_ DI dry method process were implemented. The carbon contents of Exp-E were diminished as measured by time-of-flight secondary ion mass spectrometry (TOF-SIMS), as shown in Figure 8a. The experiment confirmed that oxide voltage decreased in Test-E compared to the Exp-D, as indicated in Figure 8b.

Carbon-free O_3_ DI prevents the oxide film from being deposited uniformly through SiC formation by carbon. SiC roughens the Si surface, which affects the MOSFET characteristics leading to a degradation of the dielectric breakdown and reductions in channel mobility [15]. Using O_3_ DI, it is possible to create a uniform oxide layer to overcome the roughened surface while removing the sacrificial film by using HF. As a result, the TDDB failure rate decreased by 7 % thanks to the new process when compared to the IPA, as shown in Figure 9.

## 4. Conclusions

In this study, the pre-cleaning process factors and generation mechanisms that cause gate oxide reliability failure were identified, and methods to overcome them were investigated. After HF cleaning, it was confirmed that the concentration of IPA, which forms a direct trap site inside gate oxide, affects the decrease in G_ox_ reliability. As the concentration of IPA increases, the concentration of carbon at the Si interface increases; then in the gate oxide, carbon-based defect components form a percolation path resulting in a breakdown. To overcome this problem, a new pre-cleaning process was proposed using O_3_-DI water that replaced IPA to fundamentally solve the problem of organic contamination caused by IPA, which is the main cause of the decrease in G_ox_ reliability.

## Figures and Tables

**Figure 1 nanomaterials-12-01563-f001:**
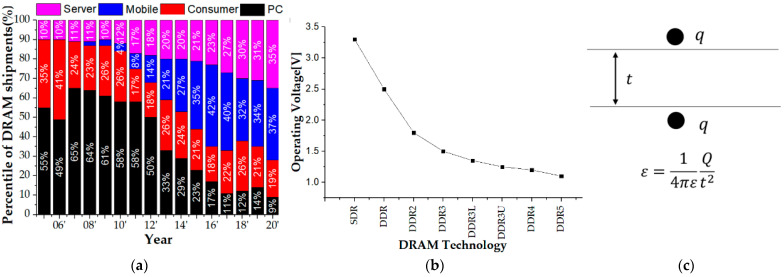
(**a**) Changes in DRAM market demand, (**b**) DRAM operating voltage history, and (**c**) electric field equation between two charges.

**Figure 2 nanomaterials-12-01563-f002:**
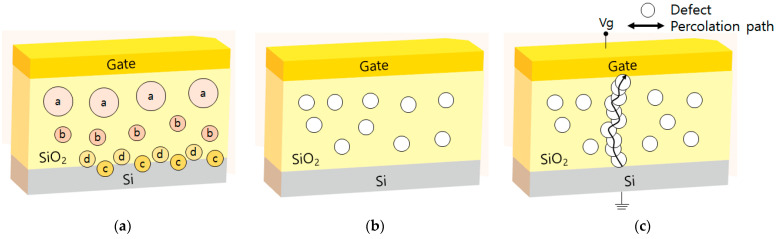
(**a**) Properties and positions of oxide charges in MOSFET structure. Percolation path (**b**) before and (**c**) after, applying the external voltage.

**Figure 3 nanomaterials-12-01563-f003:**
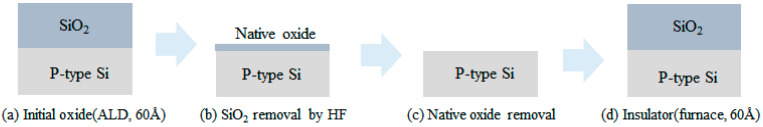
The structure of TFT process used in this study: (**a**) SiO_2_ deposited by ALD on silicon (**b**) Removal of SiO_2_ by HF cleaning (**c**) Native oxide removed by cleaning process (**d**) Gate oxide grown by vertical-type furnace on silicon.

**Figure 4 nanomaterials-12-01563-f004:**
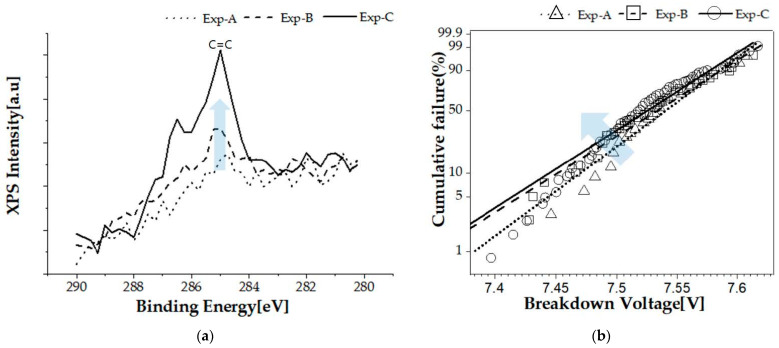
(**a**) XPS spectra of carbon and (**b**) Weibull distribution curves for breakdown voltage.

**Figure 5 nanomaterials-12-01563-f005:**
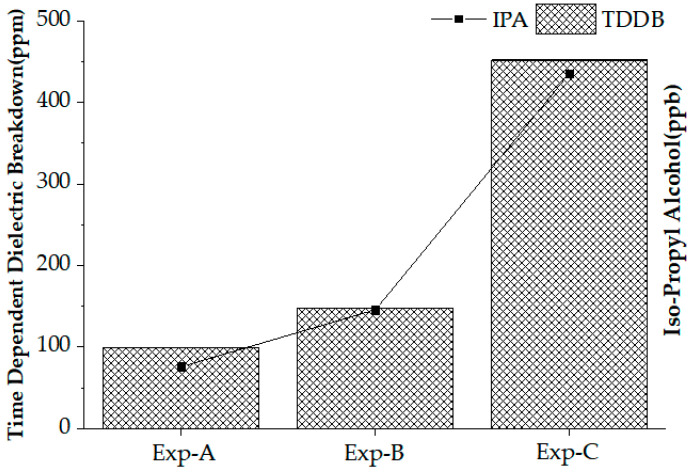
TDDB failure and IPA concentration.

**Figure 6 nanomaterials-12-01563-f006:**
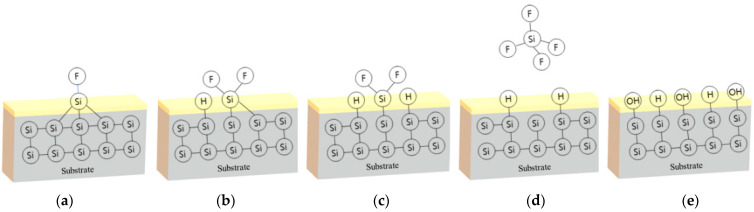
Initial oxide removal and H, OH-terminated silicon surface characteristics: (**a**) F-termination state of silicon surface (**b**,**c**) Breaking of Si–Si bonds by hydrogen (**d**) H-termination state of silicon surface (**e**) Final state of silicon surface.

**Figure 7 nanomaterials-12-01563-f007:**
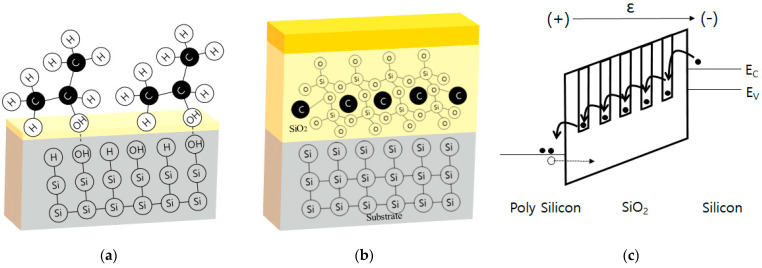
(**a**) Binding of IPA molecules at the OH-terminated interface after IPA drying process. (**b**) Bonding with carbon in Si–O Structure. (**c**) Electron-Hole pair generation (EHP) and hole trapping in SiO_2_ caused by carbon particles.

**Figure 8 nanomaterials-12-01563-f008:**
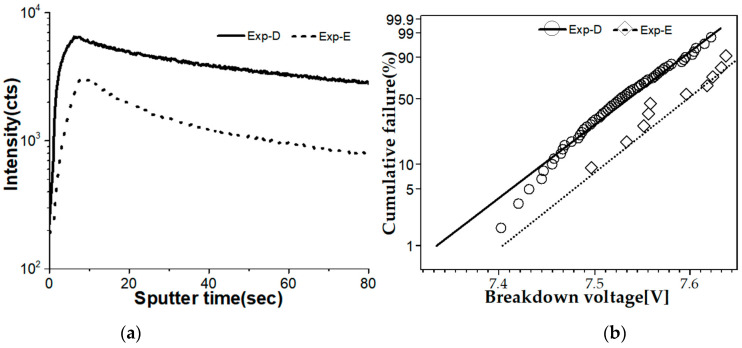
(**a**) Carbon counts of TOF-SIMS and (**b**) Weibull distribution curves for breakdown voltage for Exp-D and E.

**Figure 9 nanomaterials-12-01563-f009:**
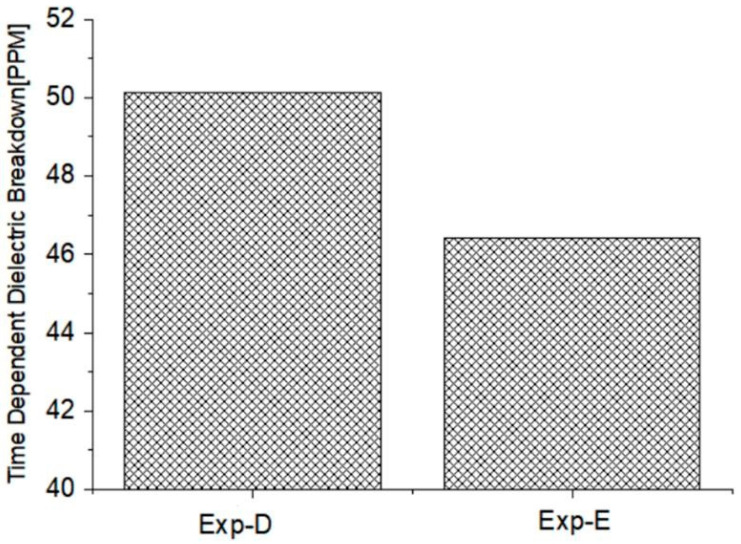
TDDB failure.

**Table 1 nanomaterials-12-01563-t001:** Weibull parameters for breakdown voltage in Figure 4b.

Sample	Scale Parameter (V)	Shape Parameter
Test-A	7.554	201.54
Test-B	7.549	172.04
Test-C	7.541	175.07

## Data Availability

The raw data required cannot be shared at this time as these data are part of an ongoing study.

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
