# Peer review of "A Study on the Role of Pre-Cleaning and a New Method to Strengthen Gate Oxide Quality"

_nanomaterials, 2022, doi:10.3390/nano12091563_

Round 1

Reviewer 1 Report

In this manuscript, the authors describe a study to understand the effect of using using IPA on the dielectric breakdown properties of MOSFETs. This is a very badly written manuscript, with lots of typos and mislabels. The language is so bad that it is hard to judge if this paper merits enough novel technical content or be published. The reviewer is not sure if this study has any relevance since current technology does not solely use SiO2. However, the topic appears to be interesting and may be useful to some people. Hence, I would strongly encourage the authors to properly address the questions below before I can consider this manuscript further.

  1. Current technology have moved from a purely SiO2 based gate dielectric and moved to a gate stack of various insulators instead. Hence, the reviewer questions the usefulness of this study since this is done on SiO2 only. Could the authors motivate the importance of this study in more details?
  2. Why did the authors pick DRAM particular? MOSFETs are not even used for charge storage in DRAMs. Why not use some examples like logic or even NAND where MOSFETs are critical?
  3. What is the purpose of the initial ALD deposited SiO2? I don’t see why did the authors need to start the experiment with an ALD deposited SiO2.
  4. On page 3, line 108 the authors write that experiment A has an IPA concentration of 10.8, experiment B has an IPA concentration of 5.4 etc. This appears to be opposite to what the authors state in the rest of the manuscript. I am confused as to which one is correct. Could u please make it consistent!
  5. How did the authors verify the oxide thickness? Please include this detail in the revised manuscript.
  6. On figure 4a, can you elaborate on which XPS peak you expect to see a change and also possibly label it clearly? The trend that the authors state appears to be opposite to what the authors stated earlier in the manuscript. Please correct these.
  7. There are lots of typos on page 5, making to impossible for the reviewer to make send of anything the authors are saying. Please update the manuscript with proper consistent language.
  8. On figure 7, IPA formula is wrong. Please fix.
  9. Please provide more details about exp D and E.

Reviewer 2 Report

In this work, the authors study the role of pre-cleaning methods on gate oxide reliability. I find this work interesting and valuable for researchers and engineers working on the technology of nanodevices. However, I find some shortcomings that need to be addressed:

  • In Section '2. Experiments and Methods’ some essential information is missing. Please indicate what tools were used. How many structures were fabricated to collect the data?
  • Add some information inf figures with XPS spectra. Mark some relevant peaks/features, it will be more understandable, clear, and informative.
  • Starting from line 127, you write about reliability. You mention some voltage values applied to the gate and no voltage to drain. Did you measure transistors? What are the dimensions? In Section 2. you did not mention anything about transistor fabrication. Please describe clearly what exactly you measured and how you did it. Why did you use only one voltage value with specific pulse intervals? Why you choose 50 uA value as an indicator of degradation? How many devices were tested to plot the figures?
  • Some simple analysis regarding the Weibull distribution would be beneficial for readers to understand the data. Add formulas describing the distribution and extract the relevant parameters of the distribution. Please also add some references to relevant publications.
  • In line 175, you mentioned TOF-SIMS measurements. Why? In Fig. 8a, we observe XPS spectra.
  • In general, authors need to clearly motivate the work. There are not many references to publications regarding the scope of the work. Most of them are not recent publications. Moreover, the authors claim that the presented method is new. There are some works on the application DI-O3 or O3 in semiconductor technology, e.g.
    • https://www.naura-akrion.com/the-application-of-di-o3-water-on-wafer-surface-preparation/
    • https://iopscience.iop.org/article/10.1149/1.1385820

Cite relevant publications and highlight the novelty of your work.

The work in this form cannot be accepted. However, I believe that after substantial improvements can be re-submitted.

Reviewer 3 Report

In this article, the author investigate how carbon is generated on the silicon interface after IPA drying process and induce the breakdown of the gate oxide. And a new method of replacing Isopropyl alcohol (IPA) with carbonless O3 deionized water (O3 DI) is proposed to improve oxide dielectric breakdown. However, there are many drawbacks needs to be solved which are listed below.

Hence, I can’t recommend that it be published in its present manuscript. Perhaps after a major revision.

  1. The Exp-A, B and C described in line 107-108 are inconsistent with the results in Fig. 4 and 5.
  2. The author should mark the carbon XPS peak and the binding energy in the XPS spectra.
  3. In lines 175-176, the author writes “The carbon contents of Exp-E were diminished as measured by time-of-flight secondary ion mass spectrometry (TOF-SIMS) shown in Fig. 8(a).”. However, Fig. 8(a) shows the XPS spectra of Exp-D and E.
  4. The caption of Fig. 8(a) does not fully describe what Fig. 8(a) shows.
  5. 8(b) lack of a caption.
  6. The Exp-D and E described in line 182 is inconsistent with the Fig. 9.
  7. Since there are some other gases that do not contain carbon,why did the author choose ozone-deionized water (O3 DI) to replace Isopropyl alcohol (IPA) in the drying process? What are the unique advantages of ozone-deionized water (O3 DI)? The author needs further description.
  8. Since Exp-A, B, C and D were both dried by IPA, why didn't the author select one from Exp-A, B and C to compare with Exp-E?
  9. The author should polish the article. And there are some errors in the article.

Examples: In line 25, the punctuation (,) or (.) are missing after the word “years”

In line 81, a period (.) is absent after the word “formed”.

In line 93, the gate oxide cannot be called a crystalline structure.

In line 143, the "I" in "SIF" should be lowercase.

Word repetition occurs in Line155, Line157, and Line161.

In reference 2, the parentheses are repeated.

Round 2

Reviewer 2 Report

The authors answered some of my questions, but they have not included all the needed information in the manuscript. 
-    Please indicate in the manuscript how many structures were fabricated to collect the data.
-    In Fig. 4a, the authors did not include any new information about relevant peaks. Please label it clearly. 
-    Authors did not answer fully the question regarding the electrical measurement. Why did you use only one voltage value with specific pulse intervals? Why did you choose the 500 uA value as an indicator of degradation? I think it is important to emphasize.
-    Weibull plot formula was added but there is no analysis and parameter extraction described regarding the presented figures. Add some relevant publications.
-    Still, figure 8a presents the XPS spectra, not the TOF SIMS data.
English needs improvement. Changes in the manuscript are written in a messy way. In general, the manuscript needs to be improved with proper language. It is hard to follow the authors reading the paper. The work in this form cannot be accepted.
